# Sensing Technology Applications in the Mining Industry—A Systematic Review

**DOI:** 10.3390/ijerph19042334

**Published:** 2022-02-18

**Authors:** Joana Duarte, Fernanda Rodrigues, Jacqueline Castelo Branco

**Affiliations:** 1Associated Laboratory for Energy, Transports and Aeronautics (PROA/LAETA), Faculty of Engineering, University of Porto, 4200-465 Porto, Portugal; jasduarte@fe.up.pt; 2RISCO, Civil Engineering Department, University of Aveiro, 3810-193 Aveiro, Portugal; mfrodrigues@ua.pt

**Keywords:** industry 4.0, zigbee network, wireless network system, safety, sensors

## Abstract

Introduction Industry 4.0 has enhanced technological development in all fields. Currently, one can analyse, treat, and model completely different variables in real time; these include production, environmental, and occupational variables. Resultingly, there has been a significant improvement in the quality of life of workers, the environment, and in businesses in general, encouraging the implementation of continuous improvement measures. However, it is not entirely clear how the mining industry is evolving alongside this industrial evolution. With this in mind, this systematic review aimed to find sensing technology applications within this sector, in order to assist the mining industry in its goal to evolve digitally. Methodology: The research and reporting of this article were carried out by means of the Preferred Reporting Items for Systematic Reviews and Meta-Analyses (PRISMA) guidelines. Results and discussion: A total of 29 papers were included in the study, with sensors being applied in several fields, namely safety, management, and localisation. Three different implementation phases were identified regarding its execution: prototype, trial, and (already) implemented. The overall results highlighted that many mechanisms are in need of improvement in underground settings. This might be due to the fact that underground mining has particular safety challenges. Conclusions: Ventilation and mapping are primary issues to be solved in the underground setting. With regard to the surface setting, the focus is directed toward slope stability and ways of improving it regarding monitoring and prevention. The literature screening revealed a tendency in these systems to keep advancing in technologically, becoming increasingly more intelligent. In the near future, it is expected that a more technologically advanced mining industry will arise, and this will be created and sustained by the optimisation of processes, equipment, and work practices, in order to improve both the quality of life of people and the health of the environment.

## 1. Introduction

Industry 4.0 has technologically revolutionised the way businesses operate and position themselves in the market. It has introduced complex problems that can only be solved through advanced sensing solutions [1].

This new industrial revolution brings the notions of digitalisation, communication, and network to light in a most complex way, moving from the traditional paper approach to a real-time, online, sharing of information [2]. Human beings, robots, machinery, and smart-gadgets work together towards the same aim: maximum efficiency in productivity.

The general framework of Industry 4.0 is usually composed of four layers: physical (devices), network (related to connectivity), big data (how the data is stored), and (final) application [3]. Sensors have many potential industrial applications in a wide range of fields: biomedical [4], operation and maintenance in civil construction [5], reinforcement of building structures [6], air quality monitoring due to traffic [7] or its noise [8], bridge structural monitoring [9], railway infrastructure monitoring [10], mine shaft monitoring [11], mine underground monitoring [12,13], risk management [14], and several other industrial applications related to productivity and product improvement [1]. 

Wireless network systems (WNS) are among the most popular means of achieving the potential and objective of the first layer because they are incredibly flexible, they do not require wiring structures, and, in most cases, they represent a low-cost alternative [3]. Wireless can be achieved through the use of WiFi, Bluetooth, WiMax, or Ultra-Wide Band technology [15]. Additionally, although they are more expensive, fibre optic sensors (FOS) improve long-distance transmission, widening the coverage area [16]. FOS do not suffer electromagnetic interference, and they have a light weight [6]. However, this type of sensor is predominantly used to measure cable strain; therefore, its application is not very broad, as it is mainly used in structural deformation monitoring [17]. These two examples stress the importance of properly suiting the sensing technology to the working objective and setting conditions. 

Mines and quarries are dynamic sites that keep evolving, with particular technical challenges. Despite all the technological enhancement, these environments are still hazardous for workers, especially in underground settings [18,19,20], where, in addition to common hazards, one can find high temperatures, humidity, gas concentration, low visibility, and narrow working areas [13,19,21]. Many challenges are born from these conditions. In the case of an accident, it is of crucial importance to determine where the miners are in order to rescue them [18]. However, whenever an incident, such as a flood or a roof collapse, a power cut is expected to occur. In this case, the traditional sensors may fail to accomplish their task, and FOS appears to be the more suitable option [16]. Additionally, communication is not always straightforward [22]. The automatisation and automation needed to solve these issues, in addition to safer work spaces, will also contribute to more environmentally friendly exploitation [21]. 

Industry 4.0 technologies have made it possible to relate variables of entirely different origins, such as those related to the production process, and those related to environmental and occupational concerns, in real time. The creation of scenarios resulting from modelling has enabled industry to opt for more productive, sustainable, solutions which do not neglect occupational needs. In this respect, the extractive industry, which follows the development, will need to make meaningful changes in order to make working conditions and practices safer. This is the anticipated technological evolution of the near future. Employees will no longer be a cost but an investment, and the industry will have to readapt once again.

Despite all the possibilities that this branch of investigation opens, it is important to remember the number of data generated by these technologies. Questions pertaining to data storage and, more importantly, data management are paramount. Artificial intelligence and machine learning may have the answers [21,23], but this is not the focus of this review. The objective of this systematic review was to find evidence, within the literature, of the potential applications of sensing technology within the mining industry, and, additionally, to determine its current applications.

## 2. Materials and Methods

The conducted research followed the Preferred Reporting Items for Systematic Reviews and Meta-analyses (PRISMA) Guidelines [24,25]. The first step consisted of selecting adequate keywords and databases to perform the search. The keywords were divided into two main groups: the first group was related to sensors, including the keywords “sensor”, “cyber physical”, “internet of things” and “virtual reality”; and the second group was related to the interest field, including the keywords “open pit”, “open cast”, “extractive industry”, “quarry”, and “underground mine”. As the initial scope of the systematic review was related to equipment, the keywords “heavy earth moving machines” and “mining equipment” were also used. The groups were sequentially combined in Scopus, INSPEC, Science Direct, and Web of Science by searching in “Title, Abstract or Keywords” fields, using the Boolean operator “AND” between sets of keywords. Following this process, exclusion criteria were defined in order to help filter the information for the first phase: (1) date—only papers published between 2017 and 2021 were included; (2) type of document—only research articles were included; (3) type of source—only peer-reviewed journals were considered; and (4) language—only articles written in English were included in the study. In the second research stage, all of the studies before 2017 were assessed, as proposed in the snowballing technique [26].

The applied eligibility criteria were related to setting: any article providing information in any environment other than real context (underground or surface mining) was immediately excluded. Exceptions were made whenever a study trialled its methodology in an actual environment. The implementation status of the system was also filtered: prototypes were included as long as they performed at least one real context trial. None of the articles were excluded due to lack of information, as long as sensor type, collected information, or system setup details were fully presented and justified. An Excel table was built to help with this task, which included the following elements: country, context, setting type, sub-process, sensing field, sensor type, connectivity, collected information, system setup, implementation, and additional information. 

The risk of bias across articles was assessed using The Cochrane Collaboration tool to assess bias [27] adapted to the systematic review scope (engineering field). This tool divides the bias into three categories: “unclear”, “low”, and “high”; it always considers the hypothesis of the different studied parameters to have an influence (“notable effect”) on the results. The categories analysed were as follows: in the methodology—equipment type, and standard application; in the results—serve the purpose, and sensor precision. Reporting quality and references quality were also assessed. 

The search and screening were conducted by one researcher, and this process was confirmed by a second researcher. Three independent researchers analysed the extracted data. 

The research was performed in July 2021.

## 3. Results and Discussion

### 3.1. Research Results

In the identification phase, 2002 records were judged against the exclusion criteria. The reasons for excluding, and the individual results, are as follows: (1) 1186 were outside the period of reference; (2) 385were excluded regarding document type; (3) 11 were excluded due to source type; and (4) 60 were excluded with regards to language. After checking both title and abstract to determine which studies were relevant, 239 more were removed. A total of 121 records were saved and uploaded to the reference manager software, Mendeley. This manager has an automatic tool that allows for the removal of all duplicate articles; a total of 39 records in this study. In the first phase, 82 records moved forward to the eligibility step. After contacting their respective authors, five records had to be removed from the results poll due to full-text unavailability. In addition to these five studies, 53 more were excluded from the research due to the following reasons: some were only reviews (that, somehow, bypassed the filters), some did not comply with the main topic (did not refer to any sensing technology), some were outside the mining scope, some only focused on algorithms or theoretical models, and some referred to un-trialled prototypes (despite being left out of the actual research results, these prototypes will receive special attention in Section 3.3). This process resulted in the inclusion of 24 records. In the next step, the snowballing technique was applied [26]: all of the references from the 24 final papers were screened in order to find new records of relevance. Regarding this stage, none of the prior exclusion criteria was used; the focus was on the set of eligibility criteria. From this process, five more papers were included in the research. Figure 1 summarises the research process, including the four different search processes, and the excluded records, per phase. In this figure, “Other = 0” refers to additional filters that could be used in the screening process. 

### 3.2. Prior Analysis

The first step was to analyse the country of origin for each study, as is displayed in Figure 2. The most represented country was China, with five studies [28,29,30,31,32], followed by Italy [33,34,35] and Korea [36,37,38], with three studies each.

The other represented nations were as follows. In Europe, Germany, Greece, Spain, Poland and Switzerland. In the Americas, Brazil, Chile in the South and the United States of America. In Asia, India, Vietnam, Israel and Hong Kong (the inclusion of Hong Kong is justified because it is an independent Chinese region).

In the next step, each study was classified according to its context (mine or quarry) and setting type (underground or surface). Figure 3 portrays a large discrepancy with relation to context, as the vast majority of the studies took place or tested their protocols in a mine setting (22 out of the 29 included articles). The reasons for this difference are not clear nor can they be determined by objective methods. However, a hypothesis is put forth here: mining commodities require more complex processes, both in terms of exploitation and processing. In its turn, it leads to the natural search for more sophisticated answers. This cannot be disassociated from the fact that 12 out of the 22 articles (which took place in a mine context) occurred in an underground setting, and this raises other exploitation concerns (especially concerning safety).

The pinpointed problems in the underground environments were as follows: deformation tunnel monitoring [39], performance of underground mapping [40], measurement of transport times [38] or improvement of loading-transport efficiency [32], and monitoring air quality [36,41]. In the study concerning the underground quarry, the main focus was on the creation of rock mass models [35]. As for the quarry surface works, the authors’ aimed to study the impact of the exploitative activities, in the vicinity of the quarry, in terms of air quality [42], mapping surface areas to plan recovery [43] or merely to comprehend the mineral distribution [44], investigating the originators of rock mass failures [33], and studying the vibrations induced by blasting [45]. Regarding surface mines, other queries emerged relative to: detection of land changes caused by mining activities [46,47], restoration monitoring [48], air quality monitoring [49], and two studies used the sensing systems to build systems—one to monitor disaster [31] and the other to monitor goaf stress [29].

### 3.3. Sensing Technology

The eligible articles were also classified by sub-process (Figure 4) and the sensing field (Figure 5) by analysing their comprehensive data in comparison to the research objective and implemented methodology. While a considerable number of studies did not report, nor did they contain any particular sub-process associated with their research, the studies that included these processes were focused on ventilation-related matters, such as the monitoring of structural issues that would compromise ventilation [39], and the development of systems that help monitor air quality [36,41,50].

Exploitation was also a topic of interest for the authors focusing on mapping structures [35,51,52]. The three studies which were related to exploration were also using sensing technologies to build and map structures [40,44,53]. Notably, it was possible to infer (whenever it was not specified) the sensing field of each article, as represented in Figure 5. 

Independent of research objectives, a sensors’ purpose is to collect information and, therefore, monitor. The path that each study followed after this process dictated the classifications provided here: safety, prediction, mapping, management, maintenance, or localisation. It is not surprising that 10 out of the 29 included papers only existed in the first complexity layer to monitor. 

This information alone does not provide much insight into the sensors’ applicability and potentiality. Information that addresses sensor data in terms of type, connectivity, collected data, system setup, and implementation status can be found in Table 1.

Although almost every system setup was sustained by a wireless network, the used terminology remained the same as the original papers in order to build this table, with captions such as “Bluetooth” or “infrared”, for example. Whenever possible, the type of collected information was registered. Despite the low application cost, this type of connectivity can have some implementation challenges, such as signal interference (and path loss), limited energy drive, and physical collision or barriers [3]; these become even more critical when mining severe environments. In this sense, Zigbee networks can offer an alternative, overcoming some of these issues, as these networks operate with low power consumption, and the nodes can communicate between themselves, despite their low data rate [20]. Nevertheless, it is important to refer to other issues associated with the use of these networks, such as the uncontrollable parameters listed here: network congestion, failures in data reception, and the number of hops; however, these parameters become somehow controllable in confined spaces [57]. 

When comparing the different sets of data, one comes across this finding: articles that focused on mapping (something) at a surface context usually included unmanned aerial systems (UAS), collecting hyperspectral images. The known applications for these systems, pertaining to mapping, traditionally revolve around slope monitoring and modelling, structure analysis, and risk assessment [58,59,60]. However, the concept and potentiality of UAS are not new, as several potentialities are documented in the literature [61]. These technologies are of great importance in all fields where access is complex, and where there is a need to collect information for three-dimensional model building [62]. The applications of UAS are vast, ranging from agriculture to military and civil engineering [63,64], and this is due to their flexibility, accessibility, low cost, and safety criteria. However, the unmanned aerial systems are not solely restricted to surface applications in the extractive sector. They can also be applied underground, particularly in harsh conditions, in order to minimise personnel exposition in addition to the improvement of exploration and exploitation [21,65,66]. This autonomation can also reduce labour intensity for the exposed workers [67].

Mining environments are complex, especially in underground operations [12,16,20,68]. Accident risks can compromise personnel as well as the mine structure itself [22]. In addition to this, communication is more complex than surface environments [15]. Most of the hazards are related to gas, such as ignition and explosion, and the contamination of the airflow [69]. The studies related to air quality monitoring (underground ventilation) which were included in this systematic review used sensors that measured gas concentrations (carbon monoxide, carbon dioxide, hydrogen sulfide, and sulfur dioxide, among others), which can be combined with temperature sensors and other parameters such as air pressure [57]. Although the gas monitorisation concept is not new, the solutions found in these studies may be. A research team is developing a helmet (still in progress) that can monitor environmental parameters and warn workers by producing alarms and light indicators concerning each parameter [68]. In addition to the novelty of such gear, this particular technology would overcome the problem of the illiteracy and/or poor technical knowledge of some workers, which adds an additional layer to training. 

Brillouin scattering, trialled in the study of Naruse et al. [39], shows great promise in the fields of geotechnical and civil engineering because of its extended range. The Brillouin Optical Time Domain Reflectometer, used in the study, has a sensing range between 20 and 50 kilometres, with a special resolution of approximately 1 metre [70], and it is highly recommended when studying temperature or strain. The results showed that this system was able to quantitatively detect the tunnel deformations caused by the imbalanced stress distribution. 

In relation to the implementation status present in the last column of Table 1, the classification was made as follows: Implemented: existing technology that used a traditional approach to solve a specific problem;Trial: existing technology using a novel approach to solve a particular problem;Prototype: technology developed by the authors, laboratory tested and validated, and, at a minimum, tested in the real ground.

Seven out of the twenty-nine analysed papers included prototypes tested in an underground environment. Three of those were in the scope of ventilation [30,41,50]. In the study of Nikolakis et al. [50], the developed prototype tried to solve a ventilation-on-demand problem by collecting gas concentrations through the sensing layer; the concentrations were compared with regulations, and the motors were adjusted in accordance. Ziętek et al. [41] developed a portable monitoring system using a smartphone, and they successfully tested it for carbon monoxide, hydrogen sulfide, temperature, and humidity. As the system structure only needs sensors, a microcontroller, and the smartphone, the authors intend to expand it to monitor other environmental parameters. Zhang et al. [30] effectively tested a radio frequency identification prototype in order to detect and predict methane concentrations, which is one of the biggest safety concerns coal mining [16,71].

Gosh et al. [40] developed a robotic prototype to map an underground structure. The generated map was then compared to the actual data, and a validation exercise was performed to check its accuracy, showing encouraging results. These developments show great promise in terms of workers’ safety, as they eliminate the inherent risk of exploring unknown areas, and, additionally, it is economically beneficial because it expands the companies’ options. In the exploring phase, these technologies can better determine whether it is feasible to mine, and, in the exploitation itself, they enable mining activities in inaccessible places for human beings. Further technologies have been developed by Li & Liu [54] to improve safety conditions. Their structure-aware self-adaptive (SASA) sensor system allows for the detection of structural variations that could cause, or be a consequence of, underground collapses. The SASA system is composed of sensors, in a mesh, that sends beacon signals to each other with location information, and this is possible because this intelligent system is able to detect any alteration inside the grid. 

Kim et al. [37] developed a smart helmet that sends proximity warning signals to pedestrians, workers, and equipment operators, which is a considerable improvement regarding safety [72]. Heavy machinery has a “blind spot” zone that can be fatal to workers on foot, and this danger can be easily reduced through proximity warning signs or alert systems [2]. An equipment proximity warning signal has also been tested in other works [32,73], adding a production management function as the navigation system [32]. In fact, the problem of fleet connectivity and management has been investigated in other studies [2,74], as compared to what is achieved in similar civil construction fields [75].

Current sensing technology does not end at these examples. Several systems are being developed every day with different purposes, and some of these purposes are listed as follows: optimal haulage system through measuring truck travel time [76], miners position for rescue in case of an accident [18], road condition monitoring [77], rock sizing, [78], hoist control [79], and, regarding processing, autonomous equipment [19,80,81].

Regarding the risk of bias (based on the methodology proposed by Higgins et al. [27]), a summary is provided in Table 2. The impact of each parameter in the results was assessed including, in its basis, the full assessment of each record. Whenever a study provided a complete task description, and definition, that could be reproducible, while presenting no notable impact, it was classified as low risk. If the article failed to provide such elements, its classification would rise to a high risk of bias. The same reasoning was applied to every other parameter. Whenever an unclear risk was used on the table, it was mainly because the original paper omitted (or simply did not use) that field in its report. Consequently, its influence in the results report was not possible to determine. The standard application was the parameter that raised more doubts: no study mentioned any protocol, guidelines, norms, or standards related to equipment or even methodology. Reference quality was also a sensitive matter due to its subjectivity: it is difficult to determine how many references are considered adequate for such protocols, and it is additionally difficult to ascertain the best or most fitted references in this regard.

### 3.4. Study Limitations

The study limitations are difficult to determine, because they are related to the systematic research. As the PRISMA statement [24,25] relies on a specific methodology, one that is not always followed by the published papers (particularly in the engineering field), the pool of results may be reduced, and these results may not always be directly relatable to the matter of interest. This can be partially overcome by including other keywords than the first set; however, this does not entirely solve the problem. Nonetheless, the authors are aware that the primary draw of the research will limit the results, and the included papers should strictly comprise the eligibility criteria. Additional records can be added to the study, but these can potentially be biased. 

## 4. Conclusions

Industry 4.0 has completely revolutionised the way in which companies think about their day-to-day operations, and, additionally, it has transformed how these businesses address the different matters ahead of them. In this sense, sensors are a cheap and efficient way of achieving all kinds of desirable results, including: industrial process improvement, product improvement, equipment/machinery monitoring, quality control, employee productivity improvement, and employee health and safety. The mining industry is not immune to these issues; on the contrary, its technical issues are more complex due to its dynamic nature, and this is one of the most hazardous economic sectors. In order to minimise or, at a minimum, mitigate the accident risk and improve workers’ safety, these sites should be monitored continuously. Another, economic, advantage of sensing technology is productivity improvement, which, ultimately, will reduce cost and increase profit. The objective of this systematic review was to find evidence of potential sensing technology applications in this field as well as its current applications. The application of the Preferred Reporting Items for Systematic Reviews and Meta-Analyses (PRISMA) led to the inclusion of 29 papers in the study. The different sensing fields across papers can be condensed in localisation, maintenance, management, mapping, prediction, safety, and, whenever it was not provided with any particular rationale to the applied protocol, monitoring. The tested solutions displayed different levels of implementation, including prototype, trial or implemented, and they were found across various sub-processes such as blasting, environment, exploitation, exploration, production, recovery, transport and ventilation. All of the abovementioned classifications were performed by this systematic review’s authors. Different solutions were found and summarised in Table 1, providing a general idea of the problems as well as the fittest protocols. Overall, the results showed a high solicitude related to underground ventilation and mapping. This may be due to the particular challenges pertaining to underground settings: gas concentration (particularly methane in coal exploitation), high temperatures and humidity, and limited working space. Overall results displayed a great concern related to safety. In the literature, two reference localisation systems were found, with technologies providing warning signs for pedestrians, workers, and machine operators [2,37]. Additionally, other fields are being tested, such as haulage optimisation systems, road condition monitoring, and autonomous equipment, to mention a few. The tendency shows that these systems will keep advancing in technological development, becoming increasingly smarter, while including machine learning and Kalman filters in order to manage information. Despite the industry’s efforts to keep up with technological development, there is still much work to be done. The mining sector needs to take advantage of the tools currently available in order to create more cost-effective productions that are, at the same time, safe for workers and for the environment.

In the near future, it is expected that workers will no longer be seen as a cost, but as an investment, and, in this sense, it will once again require the industry to adapt in order to create sustainable ways of working while considering the safety of its workers as an asset.

## Figures and Tables

**Figure 1 ijerph-19-02334-f001:**
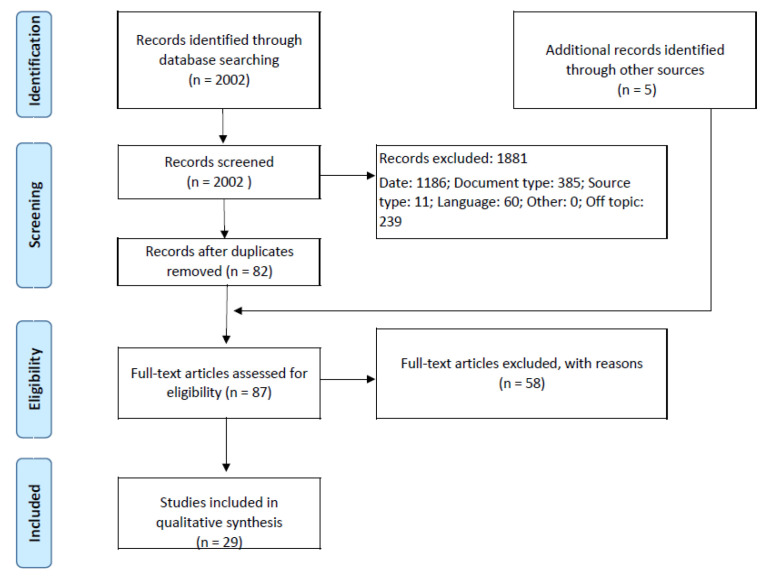
Research flow diagram.

**Figure 2 ijerph-19-02334-f002:**
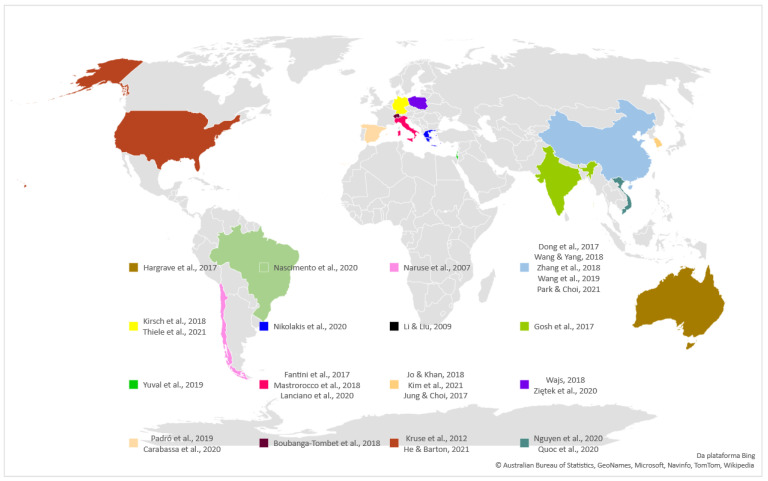
Study distribution per country.

**Figure 3 ijerph-19-02334-f003:**
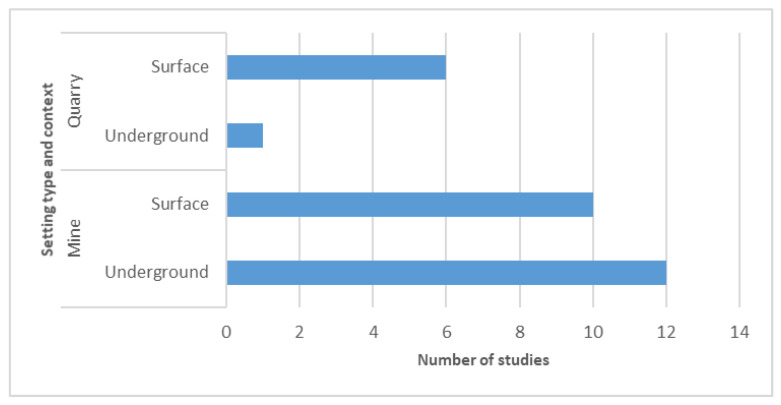
Study distribution per setting and country.

**Figure 4 ijerph-19-02334-f004:**
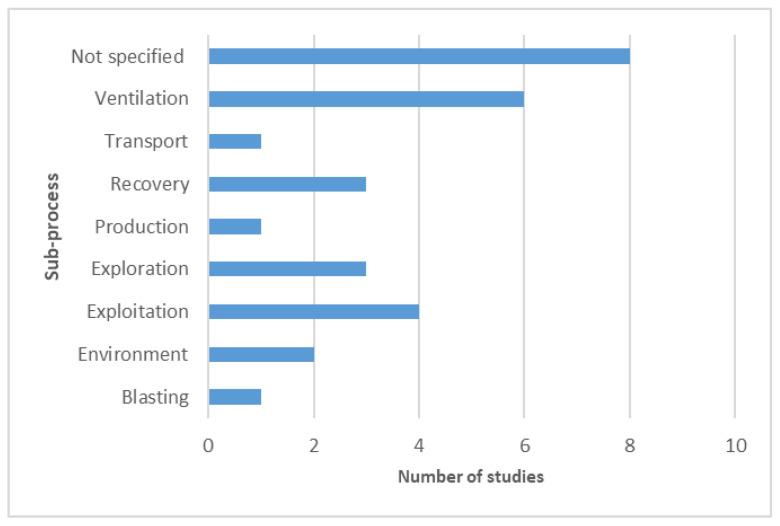
Study distribution per sub-process.

**Figure 5 ijerph-19-02334-f005:**
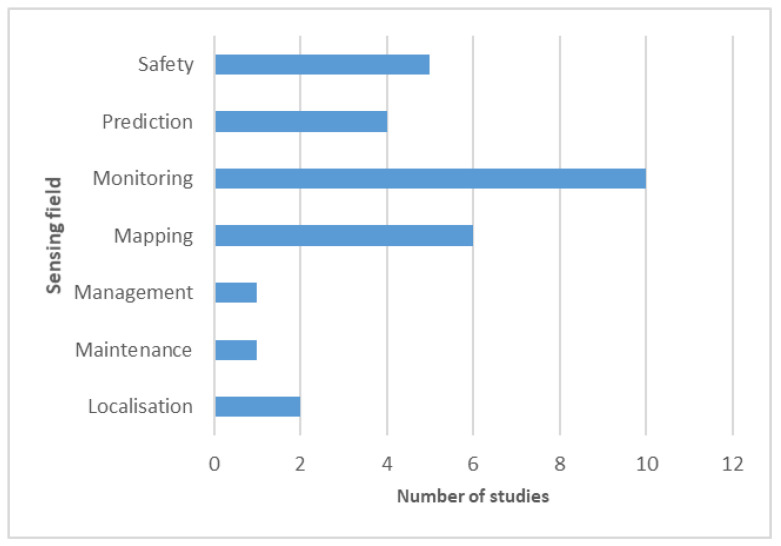
Study distribution per sensing field.

**Table 1 ijerph-19-02334-t001:** Sensors’ specifications.

Study	Sensor Type	Connectivity	Collected Information	System Setup	Implementation
[39]	Extensometer	Optical fibre	Strain distribution	Brillouin-based strain sensing system constituted by: (1) optical fibre sensors, (2) Brillouin Optical Time Domain Reflectometer (BOTDR), (3) optical switch, (4) personal computer.	Trial
[54]	Not specified	Wireless	Location	Structure-Aware Self-Adaptive (SASA) sensor system, with a beacon mechanism.	Prototype
[51]	Hyperspectral sensor, Global Positioning System, Inertial Navigation System	Infrared	Hyperspectral images, position	A Cessna 206 was equipped with a ProSpecTIR-VS scanner. The system combined a Global Positioning System (GPS), an onboard inertial navigation system (INS) and a 10 m National Elevation Dataset (NED).	Implemented
[28]	Not specified	Wireless	Vibration, pressure, temperature, noise	The system was composed of a state monitoring station, a coal mine monitoring centre, and a remote predictive maintenance system.	Not mentioned
[33]	Thermometer, hygrometer, pluviometer, anemometer, optical camera	Wireless	Wind speed and direction, rock mass temperature, strain rates	The system consisted of two weather stations, one smart camera connected to an artificial intelligence system, a stress-strain geotechnical system, one seismic monitoring device, and a nano-seismic array.	Trial
[40]	Robot	Not mentioned	(x, y, z) coordinates, yaw, pitch and roll angles	The system was composed of a robot having a differential drive mechanism.	Prototype
[55]	Radar, laser scanner	Radar	Creep position	The system consisted of one radar sensor, three laser scanners (two of them mounted broadside to the direction of travel, and the other to scan the direction of travel), and one videocamera.	Trial
[38]	Beacon	Bluetooth	Position	Bluetooth beacon system. It consists of RECO beacon and a Samsung Galaxy Note 3 smartphone.	Implemented
[44]	Thermal infrared hyperspectral sensor	Infrared	Infrared hyperspectral images (data cubes), positions	The instrument was equipped with a GPS and a high-resolution digital camera.	Implemented
[36]	Not specified	Wireless	Gases concentration, temperature	ZigBee wireless communication protocol: (1) data acquisition system, (2) data transmission, (3) data processing (quality assessment and prediction), (4) information sharing, (5) intelligent control of mine ventilators.	Implemented
[53]	Optical sensors	Infrared	Outcrop images, hyperspectral image, radiance image, hyperspectral scan	The system was composed of a hyperspectral push broom scanner (Specim AisaFenix Terrestrial)—visible to near-infrared and shortwave infrared, a hyper-cam (Telops Hyper-Cam LW)—longwave infrared, and a drone-borne (Senop Rikola)—visible to near-infrared	Implemented (new approach)
[35]	Optical sensors	Infrared	Absolute (x, y, z) coordinates of the point clouds, the reflectance (reflectivity) of surfaces, RGB data from the associated photographic images	The system consisted of a LiDAR using a pulse-based static terrestrial laser scanner. In addition, topographic data (GPS and TS) was used.	Implemented
[46]	Multispectral sensor	Infrared	Multispectral images	The system was composed of two satellites: SENTINEL-1 and SENTINEL-2.	Implemented
[31]	Surveying robot, GPS, air temperature and pressure sensor	Wireless	Air temperature, pressure, position coordinates.	Geosensor network. The system consists of a data sensing layer, a data management layer, sensor services, and an application layer.	Implemented
[30]	Methane gas density sensor	Wireless	Radiofrequency identification (RFID)	Wireless sensor network. The system consisted of a communication section, a radio-frequency front-end section, and a digital section.	Prototype
[48]	Optical sensor	Near-infrared	Multispectral images	Unmanned aerial system (UAS) with four spectral bands: green (GRE), red (RED), red-edge (REG) and near-infrared (NIR).	Implemented
[29]	Tension bar stress meters, photo-elastic stress meters, stress-strain borehole stress meter	Wireless	Stress (MPa),	ZigBee wireless network. The system consists of four parts: a stress monitoring unit, a data acquisition unit, a wireless communication unit, and a database management unit.	Trial
[42]	Optical sensor	Not mentioned	Particle number concentration (PNC)	A network of low-cost sensors composed the system.	Implemented
[43]	Optical sensor	Not mentioned	Multispectral images	Unmanned aerial system (UAS).	Trial
[34]	Optical fibre sensor	Not mentioned	Temperature, strain, images	Unmanned aerial system (UAS) consisting of the Inertial Navigation System (INS) with GNSS, accelerometers and gyroscopes, a video camera for remote inspection and the flight management software. Topographic monitoring system was composed of a laser distancemeter, an electronic theodolite, and a computer.	Trial
[47]	Orbital sensors (satellite), optical sensor	Infrared	High-resolution images	The system was composed of a satellite and a LiDAR.	Trial
[45]	Geophone sensors, vibration sensor	Not mentioned	Intensity of vibration, vibration	The system consisted of vibration sensors.	Trial
[50]	Not specified	Wireless	Gases concentration (CO_2_, CO, NO_2_, NO, O_2_, SO_2_, H_2_S), temperature, humidity	The data transmission was achieved through a set of LoRa nodes. The ventilation motor control component was implemented as web applications in Java.	Prototype
[49]	Electrochemistry sensors, laser dust sensor	Not mentioned	PM_2.5_, CO, CO_2_, and SO_2_ concentrations	Unmanned aerial system, where the sensors were attached to the drone.	Trial
[41]	CO gas sensor, H_2_S gas sensor, temperature sensor, pressure sensor, humidity sensor	Bluetooth	Gases concentration, temperature, pressure, humidity	The system consists of four modules: sensor layers, data acquisition by the microcontroller, smartphone, and external IT infrastructure on the surface (optional).	Prototype
[56]	Optical sensor	Not mentioned	Spectral images	The system consisted of a drone and tripod-mounted sensors.	Implemented
[37]	Smart helmet, Bluetooth beacon	Bluetooth low energy	Not mentioned.	The system consisted of two sensors.	Prototype
[32]	Not specified	Bluetooth	Location	The system was composed of a Bluetooth beacon and a tablet PC.	Prototype
[52]	Hyperspectral sensor, optical sensor	Not mentioned	Hyperspectral images, outcrop scans	Unmanned aerial system combined with LiDAR.	Implemented

**Table 2 ijerph-19-02334-t002:** Risk of bias.

Study	Methodology			Results		Other	
	Task Definition	Equipment Type	Standard Application	Serve the Purpose	Sensor Precision	Reporting Quality	References Quality
[39]	LR	LR	UR	LR	UR	LR	HR
[54]	LR	UR	UR	UR	LR	LR	HR
[51]	HR	LR	UR	LR	UR	LR	HR
[28]	HR	UR	UR	UR	UR	HR	HR
[33]	LR	LR	UR	LR	UR	HR	HR
[40]	LR	UR	UR	UR	UR	LR	HR
[55]	LR	LR	UR	LR	LR	LR	HR
[38]	LR	HR	UR	LR	UR	LR	HR
[44]	LR	LR	UR	LR	LR	LR	LR
[36]	LR	UR	UR	UR	LR	LR	LR
[53]	LR	LR	UR	LR	LR	LR	LR
[35]	LR	LR	UR	LR	LR	LR	LR
[46]	HR	HR	UR	LR	UR	HR	HR
[31]	HR	LR	UR	LR	LR	LR	HR
[30]	HR	UR	UR	UR	UR	LR	LR
[48]	LR	LR	UR	LR	LR	LR	LR
[29]	LR	LR	UR	LR	UR	LR	HR
[42]	LR	UR	UR	UR	UR	LR	HR
[43]	HR	LR	UR	LR	UR	LR	LR
[34]	LR	LR	UR	LR	UR	LR	LR
[47]	LR	LR	UR	LR	LR	LR	LR
[45]	LR	LR	UR	LR	UR	LR	LR
[50]	HR	UR	UR	UR	UR	HR	LR
[49]	HR	LR	UR	LR	UR	LR	LR
[41]	LR	LR	UR	LR	LR	LR	LR
[56]	LR	LR	UR	LR	LR	LR	LR
[37]	LR	LR	UR	LR	LR	LR	LR
[32]	LR	LR	UR	LR	UR	HR	HR
[52]	LR	LR	UR	LR	UR	LR	LR

HR—High Risk; LR—Low Risk; UR—Unclear Risk.

## Data Availability

Not applicable.

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
