# Peer review of "Sensing Technology Applications in the Mining Industry—A Systematic Review"

_ijerph, 2022, doi:10.3390/ijerph19042334_

Round 1

Reviewer 1 Report

The use of English is very stilted and is very hard to read and understand. However, the biggest problem that I see is "Why was the study done?" and "Who will benefit from reading the paper?" the abstract and Introduction need to clearly address these two points. You have put a lot of work into the paper so redo the intro and abstract so that a reader can tell why he or she should be reading the paper. 

Reviewer 2 Report

At first I would like to say that I like your paper very much, because it is really about the huge progress that can be made thanks to modern technologies - sensors even in such a complex area as mining. Also I would like to mention part – Materials and Methods, which are prepared very precisely. Inclusion of 29 papers in the study is quite a nice number.

What I missed in article is literature revue about the similar made (if there are some...) researches, not only in mining, but in general in this modern age in several industries and how it was in history with senzors using. Also there is very actual and interesting field – virtual reality projects (EU projects, structural funds etc.), which are dealing with opportunities of senzors using also in mining, which could make work easier, save time and money (but this is maybe something special for another separate article). 
Also I missed in the conclusion something specifically that the authors came up with in their research, because I think that the fact that the sensors are using, is currently clear.

Reviewer 3 Report

The article, presenting the state of the art of sensing technology in mining industry, concerns on a very important subject. Sensors, digital twinning and autonomous machines are the future of mining industry, operating in increasingly difficult conditions and meeting new safety and environmental restrictions. I am impressed with author’s work on the research. However, the article still may be improved:

  • Figure 1 might be of better quality.
  • Map and text on figure 2 is barely readable. Can you please make it bigger?
  • Lines 145-146: You state that “The reasons for this difference [between the number of articles about mines and quarries] are not clear nor can be determined by objective methods.”. But do you have any hypothesis? I think you can present some.
  • I am surprised that you do not write about articles concerning on sensors used in machines (hoists, LHDs etc.). As more and more autonomous machines are developed for both underground and open cast mines, I was expecting that this topic will dominate your article. What is the reason in your opinion? Is it because of the key words selection? Or did you do it on purpose? Also I was expecting some more about land surveying.
  • Table 1. - I am aware of the author's guidelines provided by the publisher, but this table is almost impossible to read. Horizontal division lines would make it much more clear.
  • Lines 212-219: Well, it is worth noting that gas sensors are in common use in mines for decades. Automatic gasometers are in service probably in all mines, where gas hazard occur.
  • Table 2. - What do you think about using colours to emphasis level of risk? Is it worth it or is it table just not important enough?
  • Lines 324-325: I think that there are some more reasons of this situation. Gas sensors and devices for geodesy are developed for decades. There are also used outside the mining industry (like civil engineering), so there are more applicable and thus more profitable to develop. What do you think about that point of view?
  • There is a room for improvement of English.
  • Consider referring to https://www.intechopen.com/online-first/75296.

Reviewer 4 Report

Dear authors:

The paper, although extensive, is very basic and the main contributions have not been highlighted.

These are the issues that can help to improve your work:

  1. The paper, although extensive, is very basic and the main contributions have not been highlighted.
  2. These are the issues that can help to improve your work:
  3. You should modify the abstract according to the input of your review on the subject.
  4. The use of FOS is not fully justified in the introduction.
  5. There is no reference to support the argument: “ The automatization and automation needed to solve these issues, on top of safer working places, will also contribute to greener exploitation”.
  6. It should be more explicit about what is novel in your paper. The last paragraph of the introduction is ambiguous.
  7. In the Materials and methods section, the keywords mentioned in the abstrat have not been used in the keyword search. There are some that are considered important. For example: Industry 4.0, Wireless network system, sensor, safety, underground or surface mining, ...
  8. You do not indicate any characteristics of the researchers who did the screening. Are they experts? How were they selected? What parameters were used to determine that they are experts in the field?
  9. The system of elimination of articles does not bring anything new, it is only the use of the filters of the databases when searching. It then explains the functionality of the reference manager, but this question is a must in any bibliographic review.
  10. The limitations of the study are not robustly justified.
  11. There are conclusions that are rather generic and should be reviewed to highlight the contributions of the study conducted. For example: “Other advantages of using 309 sensing technology to the company’s advantage are the productivity improvement, 310 which, ultimately, will directly impact the costs’ reduction and increased profit”
  12. References to other papers are usually placed in the body of the paper (introduction and background, methodology, results, or discussion) and not in the conclusions.

Yours sincerely

Round 2

Reviewer 1 Report

The per is very difficult to read because of the stilted use of language. However they have declined to edit it for clarity. 

Reviewer 4 Report

Dear authors,
Thank you for incorporating the suggestions.
Yours sincerely